Expression of microRNA-155 in thalassemic erythropoiesis

Penglong Tipparat 1
Pholngam Nuttanan 2 3
Tehyoh Nasra 4
Tansila Natta 4
Buncherd Hansuk 4
Thanapongpichat Supinya 4
Srinoun Kanitta kanitta.s@psu.ac.th 4
1 Department of Pathology, Faculty of Medicine, Prince of Songkla University , Hat Yai , Songkhla , Thailand
2 Molecular Medicine Graduate Program, Faculty of Science, Mahidol University , Bangkok , Thailand
3 Thalassemia Research Center, Institute of Molecular Biosciences, Mahidol University , Nakhon Pathom , Thailand
4 Faculty of Medical Technology, Prince of Songkla University , Hat Yai , Songkhla , Thailand
Marunaka Yoshinori
Electronic publication date: 2024 Sep 20
Publication date: 2024
Volume: 12
Electronic Location ID: e18054
Received 2024 May 2; Accepted 2024 Aug 16
Copyright: ©2024 Penglong et al.
Copyright year: 2024
Copyright holder: Penglong et al.
License: This is an open access article distributed under the terms of the Creative Commons Attribution License, which permits unrestricted use, distribution, reproduction and adaptation in any medium and for any purpose provided that it is properly attributed. For attribution, the original author(s), title, publication source (PeerJ) and either DOI or URL of the article must be cited.
License URL: https://creativecommons.org/licenses/by/4.0/

Keywords: miR-155, Ineffective erythropoiesis, Thalassemic mice, c-myc

Funding: The government budget of the Prince of Songkla University MET6402028S This work was supported by the government budget of the Prince of Songkla University under grant MET6402028S. The funders had no role in study design, data collection and analysis, decision to publish, or preparation of the manuscript.

==============================
Background

Ineffective erythropoiesis (IE) is the primary cause of anemia and associated pathologies in β-thalassemia. The characterization of IE is imbalance of erythroid proliferation and differentiation, resulting in increased erythroblast proliferation that fails to differentiate and gives rise to enucleate RBCs. MicroRNAs (miRs) are known to play important roles in hematopoiesis. miR-155 is a multifunctional molecule involved in both normal and pathological hematopoiesis, and its upregulation is observed in patients with β-thalassemia/HbE. However, the expression and function of miR-155, especially in β-thalassemia, have not yet been explored.

Methods

To study miR-155 expression in thalassemia, erythroblast subpopulations, CD45-CD71+Ter-119+ and CD45-CD71−Ter-119+ were collected from βIVSII-654 thalassemic bone marrow. Additionally, a two-phase culture of mouse bone marrow erythroid progenitor cells was performed. Expression of miR-155 and predicted mRNA target genes, c-myc, bach-1 and pu-1, were determined by quantitative reverse transcription (qRT)-polymerase chain reaction (PCR) and normalized to small nucleolar RNA (snoRNA) 202 and glyceraldehyde-3-phosphate dehydrogenase (GAPDH), respectively. To investigate the effect of miR-155 expression, erythroblasts were transfected with miR-inhibitor and -mimic in order to elevate and eliminate miR-155 expression, respectively. Erythroid cell differentiation was evaluated by Wright–Giemsa staining and flow cytometry.

Results

miR-155 was upregulated, both in vivo and in vitro, during erythropoiesis in β-thalassemic mice. Our study revealed that gain- and loss of function of miR-155 were involved in erythroid proliferation and differentiation, and augmented proliferation and differentiation of thalassemic mouse erythroblasts may be associated with miR-155 upregulation. miR-155 upregulation in β-thalassemic mice significantly increased the percentage of basophilic and polychromatic erythroblasts. Conversely, a significant decrease in percentage of basophilic and polychromatic erythroblasts was observed in β-thalassemic mice transfected with anti-miR-155 inhibitor. We also examined the mRNA targets (c-myc, bach-1 and pu-1) of miR-155, which indicated that c-myc is a valid target gene of miR-155 that regulates erythroid differentiation.

Conclusion

miR-155 regulates IE in β-thalassemia via c-myc expression controlling erythroblast proliferation and differentiation.

Introduction

β-thalassemia consists of a diverse group of inherited blood disorders caused by mutations in the β-globin gene, resulting in imbalance of α- and β-globin chains and ineffective erythropoiesis (IE), which eventually produce severe iron-loading anemia and others pathophysiology (Oikonomidou & Rivella, 2018; Schrier, 2002; Taher, Weatherall & Cappellini, 2018). IE is defined by increased proliferation and expansion of immature erythroblasts but a limited production of mature erythrocytes (Pootrakul et al., 2000; Rund & Rachmilewitz, 2005). Previous studies revealed that IE is caused by accelerated erythroid proliferation, maturation blockade, death of erythroid precursors, apoptosis, and autophagy (Chaichompoo et al., 2022a; Chaichompoo, Svasti & Smith, 2022b; Libani et al., 2008; Mathias et al., 2000; Pootrakul et al., 2000; Srinoun et al., 2009). Although, molecular defects causing β-thalassemia have been extensively studied, mechanisms regulating IE remains unclear.

MicroRNAs (miRs or miRNAs) are a class of small non-coding regulatory RNAs that post-transcriptionally regulate gene expression by binding to the 3′-untranslated regions of their target mRNAs and repressing protein production by destabilizing the mRNA and translational silencing (Ha & Kim, 2014). miRNAs play key functions in many cellular pathways, including normal erythropoiesis. Moreover, the dysregulation of miRNAs involved in erythropoiesis have been reported for erythropoietic disorders (Azzouzi, Schmugge & Speer, 2012; Lawrie, 2010; Li et al., 2023), including their role in the pathogenesis of β-thalassemia. In β-thalassemia, upregulation of miR-451 has been shown to occur in early erythropoiesis (Svasti et al., 2010), with increased level of circulating plasma miR-451 found in β-thalassemia/HbE patients, which correlated with severity of anemia (Leecharoenkiat et al., 2017). Moreover, miR-210 dysregulation was observed during hypoxia in β-thalassemia/HbE progenitor cells (Sarakul et al., 2013), with increased level of plasma miR-210 being associated with disease severity in β-thalassemia/HbE patients (Siwaponanan et al., 2016). Recently, upregulated miR-101-3p was reported to be associated with high proliferation of the erythroblasts in β-thalassemia/HbE patients (Phannasil et al., 2023). Das et al. (2021) has reported significant upregulation of eight miRNAs, including miR-192-5p, miR-335-5p, miR-7-5p, miR-98-5p, miR-146a-5p, miR-146b-5p, miR-148b-3p and miR-155-5p in β-thalassemia/HbE erythropoiesis. There may also be other miRNAs that are dysregulated in β-thalassemia erythropoiesis which need to be explored.

miR-155 has been shown to be involved in erythropoiesis, with studies reporting downregulation of miR-155 expression during the late stages of erythropoiesis (Bruchova et al., 2007; Masaki et al., 2007). Elevated miR-155 expression leads to fewer erythroid progenitors and erythrocytes during erythropoiesis (Georgantas 3rd et al., 2007; Li et al., 2023). miR-155 expression is significantly higher in myelodysplastic syndrome (MDS), an IE-related blood disease (Wan et al., 2020). Likewise, upregulation of miR-155 was observed in β-thalassemia/HbE (Das et al., 2021). Thus, dysregulation of miR-155 has been implicated in IE. Based on the above information, we hypothesized dysregulation of expression and function of miR-155 in β-thalassemia, which has not yet been explored. Therefore, we investigated the expression profiles of miR-155 in β-thalassemia by in vivo and in vitro methods. We also identified a target gene of miR-155 which may be involved in regulation of erythropoiesis in β-thalassemia.

Materials & Methods

Study animals and bone marrow specimens

Wild type (WT) (βm/m) and βIVSII-654-thalassemic mice (Lewis et al., 1998) littermates under C57BL/6J background (twelve mice per each of the group) were obtained from the Thalassemia Research Center, Institute of Molecular Biosciences, Mahidol University, Thailand. The βIVSII-654-thalassemic C57Bl/6 mouse model demonstrated comparable features to thalassemia intermedia, including anemia, ineffective erythropoiesis and splenomegaly (Srinoun et al., 2009). Male or female 8-to-12-week-old βIVSII-654-thalassemic and WT mice were housed 4–6 per cage per genotype. All twenty-four were given routine feeding with standard diet and water ad libitum. The temperature was 25 ± 2 °C and humidity was maintained at 55 ± 10%, with a 12-h light/dark cycle and clean conventional housing system. The mice were euthanized by carbon dioxide (CO2) inhalation. No exclusion criteria were used.

After the mice were euthanized, bone marrows from the femur were collected by flushing the femurs with collection media containing 2% fetal bovine serum (FBS) in Iscove’s modified Dulbecco’s medium (IMDM) (GIBCO, Waltham, MA, USA) (Srinoun et al., 2009).

The protocol of this study was approved by the Ethics Review Board of the Institute of Science and Technology for Research and Development and the Institute of Molecular Biosciences, Mahidol University, Institute Animal Care and Use Committee (approval number MUSTA 2008-004 and COA.NO.IMB-ACUC 2021/011, respectively).

Purification of erythroblasts from mouse bone marrow

Erythroblast subpopulations from mouse bone marrow from 3 WT and 3 β-thalassemic mice were isolated by magnetic cell sorting using anti-mouse CD45 immunomagnetic beads (Miltenyi Biotec, Bergen-Gladbach, Germany) to separate white blood cells from erythroid cells. The cells were then separated into different stages of erythroid differentiation according to transferrin receptor levels using CD71(transferrin receptor) immunomagnetic beads. This resulted in two erythroblast subpopulations, CD45-CD71+Ter-119+ and CD45-CD71-Ter-119+ in bone marrow. The purity and stage of erythroid differentiation were examined by staining with anti-mouse Ter-119 (red cell marker)-PE and anti-mouse CD71-fluorescein isothiocyanate (FITC) (BD Biosciences, San Jose, CA, USA) and analyzed using a FACSCalibur flow cytometer and CellQuest software (BD Biosciences).

Mouse bone marrow erythroid progenitor cell culture

Single cell suspensions of mouse bone marrow cells (WT n = 3, β-thalassemic mice n = 3) were suspended in collection media and Ter-119+ cells were depleted by negative selection using biotin-conjugated anti-mouse Ter-119 antibody (BioLegend, San Diego, CA, USA) and streptavidin microbeads (Miltenyi Biotec, Teterow, Germany). Ter-119− cells were collected using a system of MACS cell separation (Miltenyi Biotec, Teterow, Germany) according to the manufacturer’s protocol. The Ter-119 negative cells were cultured in two-phase liquid culture (d’Arqom et al., 2021). Initially, the expansion phase was performed, Ter-119 negative cells were cultured in StemPro- 34 serum-free medium (Gibco, Waltham, MA, USA), supplemented with 1x nutrient supplement mix (Gibco), 1 µM dexamethasone (Sigma-Aldrich, St. Louis, MO, USA, 2 mM L-glutamine (Gibco), 2 U/mL erythropoietin (Stemcell Technologies, Vancouver, Canada), 40 ng/mL mouse insulin-growth factor-1 (Stemcell Technologies), 180 ng/mL mouse stem cell factor (PeproTech, Rocky Hill, NJ), and 100 U/mL penicillin/ streptomycin (Gibco) at 2 × 106 cells/mL in 48-well plates and incubated at 37 °C, 5% CO2 for 5 days. Cells in the expansion phase were then cultured in conditioned media during the differentiation phase. Cells were cultured in Iscove’s modified Dulbecco’s medium (Gibco) supplemented with 15% FBS (Gibco), 2 mM L-glutamine, 2 U/mL erythropoietin, 200 µg/mL human holotransferrin (ProSpec-Tany TechnoGene Ltd., Ness-Ziona, Israel), 10 µg/mL insulin (Gibco), 1% bovine serum albumin (HiMedia Laboratories Pvt., Ltd. Maharashtra, INDIA), 0.1mM β-mercaptoethanol (Sigma-Aldrich) and, 100 U/mL penicillin/streptomycin at 37 °C, 5% CO2 for 2 days.

Erythroid differentiation analysis

Erythroid cell morphology and cell marker during differentiation was evaluated by Wright–Giemsa staining and flow cytometry, respectively. The murine erythroid cell culture was cytospun and analyzed by observing the Wright-Giemsa-stained cells under a light microscope. For flow cytometry, cultured erythroblasts were stained with anti-mouse Ter-119-PE and anti-mouse CD71-FITC antibodies (BD Biosciences, Franklin Lakes, NJ, USA). Erythroid subpopulations were identified as previously described (Chen et al., 2009; Liu et al., 2006). Data of 100,000 events was acquired and analyzed using BD Accuri™ C6 Plus Flow Cytometry (BD Biosciences) and evaluated using FlowJo software v10 (FlowJo LLC, BD Bioscience).

Transfection of miR-155 mimic and anti-miR-155 inhibitor

During the expansion phase of the culture (day 0), mouse erythroid cells (5 × 105 cells) were treated with 80 nM of either miR-155 miRNA mimic or negative mimic control (mirVana™ miRNA mimic; Applied Biosystems, Waltham, MA, USA) to increase miRNA expression (WT mice n = 3, β-thalassemic mice n = 3), or miR-155 inhibitor or negative inhibitor control (Anti-miR™ miRNA Inhibitor; Applied Biosystems) to inhibit miRNA expression (WT mice n = 3, β-thalassemic mice n = 3). Both, miRNA-mimic and -inhibitor were transfected into erythroid cells using Lipofectamine™ RNAiMAX (Invitrogen, Waltham, MA, USA). Erythroblast cells were collected on day1 and day2 (differentiation phase), and the expression of miR-155 and its mRNA target genes was analyzed using quantitative reverse transcription PCR (qRT-PCR). Erythroid cell differentiation was evaluated by flow cytometry.

Quantitative miRNA analysis by RT-qPCR

Total RNA was extracted from cells using a Hybrid-R™ miRNA Isolation Kit (Gene-All, Seoul, South Korea). The concentration of RNA was measured with a NanoDrop 2000 Spectrophotometer (Thermo Fisher Scientific, Waltham, MA, USA) with a suitable A260/280 ratio of 1.8 to 2.2, and a suitable A260/230 ratio of >1.7 and stored at −20 C. miR-155 quantitation was performed by using TaqMan™ Advanced miRNA cDNA Synthesis Kit (Thermo Fisher Scientific) and TaqMan ™ Universal Master Mix II with UNG (Thermo Fisher Scientific) according to the manufacturer’s protocol. Let7a and snoRNA202 were used as reference genes for in vivo and in vitro experiments, respectively. The expression of miRNA normalized on reference gene was calculated using 2−ΔCt (comparative Ct) method by using WT or untreated as control. miRNA quantitation was analyzed using a LightCycler® 480 PCR System (Roche Applied Science, Mannheim, Germany).

Analysis of putative mRNA expression targets of miR-155

The candidate mRNA targets of miR-155 (c-myc, bach1 and pu1) were validated using qRT-PCR. cDNA was synthesized from the total RNA and amplified in one-step qRT-PCR using the qPCRBIO SyGreen 1-step Go Lo-ROX Kit (PCR Biosystems Ltd., London, UK) with specific primers. The primers used in the qRT-PCR were as follows: c-myc: forward primer, 5′-ATGCCCCTCAACGTGAACTTC-3′; reverse primer, 5′-CCTCTTCTCCACAGACA CCAC-3′, bach1: forward primer 5′-GCCCGTATGCTTGAGTGAATT-3′, reverse primer 5′- CGTGAGAGCGAAATTATCCG-3′, pu1: forward primer, 5′-ATGTTACAGGC GTGCAAAATGG-3′, reverse primer, 5′-TGATCGCTATGGCTTTCTCCA-3′; Actb: forward primer, 5′-CGTTGACATCCGTAAAGACCTC-3′; reverse primer, 5′-AGCCACCGATCCACACAGA-3′. Firstly, reverse transcription step was performed at 45 °C for 10 min. Secondly, PCR amplification was performed with an initial denaturation at 95 °C for 2 min followed by 40 cycles of denaturation at 95 °C for 5 s, annealing and extension at 60 °C for 30 s, and melt curve stage. The experiments were performed in triplicates. The expression of mRNA target was normalized with β-actin (Actb) as an internal control and then performed using the 2−ΔΔCt method. qRT-PCR was performed to amplify mRNA using a LightCycler® 480 PCR System (Roche Molecular System).

Statistical analyses

Data were analyzed using Statistical Package for the Social Sciences version 23 (SPSS Inc., Chicago, IL, USA). Comparisons between parameters were conducted using the nonparametric Mann–Whitney U Test. Values of P <0.05 were accepted as statistically significant.

Results

In vivo and in vitro upregulation of miR-155 in β-thalassemic mice erythropoiesis

IE has been reported in β-thalassemic mice (Chaichompoo et al., 2022a; Srinoun et al., 2009). To study miR-155 expression, erythroblasts from β-thalassemic mice bone marrow was separated into two erythroblast subpopulations: CD45-CD71+Ter-119+ and CD45-CD71−Ter-119+, represent early and late erythroblasts, respectively. The purity and stage of erythroid differentiation were determined by flow cytometry and Wright–Giemsa staining, respectively. The results demonstrated that the CD45-CD71+Ter-119+ fraction mainly contain basophilic and polychromatic erythroblasts, while the CD45-CD71−Ter-119+ fraction comprised mainly of orthochromatic erythroblasts and mature erythrocyte (Figs. 1A and 1B). Our results showed increased basophilic and polychromatic erythroblasts which represented by CD45-CD71+Ter-119+ fraction in β-thalassemic mice compared to WT (Figs. 1A and 1B). The expression levels of miR-155 in CD45-CD71+Ter-119+ and CD45-CD71−Ter-119+ fractions from WT and β-thalassemic mice were determined using qRT-PCR. Interestingly, miR-155 expression level in early erythroblast was significantly lower than that of late erythroblast in β-thalassemic mice. Moreover, miR-155 expression in two erythroblast subpopulations of β-thalassemic mice were significantly upregulated compared to that of wild type mice (Fig. 1C).

Figure 1 In vivo expression of miR-155 during erythropoiesis in β-thalassemic mice.

Erythroid subpopulation of CD45-CD71+Ter-119+ and CD45-CD71-Ter-119+ cell was separated by magnetic cell sorting from mouse bone marrow. The purity and stage of erythroid differentiation were determined by flow cytometry (A) andWright-Giemsa staining (B). Quantitative RT-PCR of (C) miR-155 was determined in WT, thalassemic mice normalized by the expression of let-7a (2−ΔCt). The experiments were carried out in triplicate. *p < 0.05, **p < 0.01 by Mann–Whitney U-Test.

To investigate the expression of miR-155 during in vitro erythropoiesis, Ter-119-depleted erythroid progenitor cells isolated from the heterozygous βIVSII-654-thalassemic mice bone marrow were cultured in a two-phase liquid culture system. In the expansion phase (day 0), cell proliferation did not differ between the thalassemic and WT mice. However, in the differentiation phase, the proliferation rate in cells derived from thalassemic mice increased to approximately 1.6-fold on day1 and 1.3-fold on day2 compared with that of the WT (Fig. 2A). These results correspond to increased proliferation of β-thalassemic erythroid cells due to IE. In the expansion phase, erythroblasts exhibited high CD71 and intermediate Ter-119 expression corresponding to proerythroblasts (region S1) in both WT and thalassemic mice (Figs. 2B and 2C). Although during differentiation phase, the main population of erythroblasts comprises early erythroblasts or proerythroblasts, the cells increasingly differentiated into basophilic erythroblasts (region S2), polychromatic erythroblasts (region S3), and orthochromatic erythroblasts (region S4) on day1 and day2, respectively (Figs. 2B and 2C). On day 2, the result revealed that polychromatic erythroblasts of thalassemic mice were significantly increased compared with those of WT mice (Fig. 2C). Erythroid cell differentiation was also evaluated by Wright–Giemsa staining, which showed that erythroblast morphology corresponded with erythroid marker outcomes. Moreover, our bone marrow culture conditions demonstrated that erythropoiesis occurred in an erythroblastic island niche. It consisted of a central macrophage encompassed by erythroblasts (Fig. 2D).

Figure 2 Proliferation and differentiation of erythroblast and expression of miR-155 during in vitro erythropoiesis in β-thalassemic mice.

(A) Cell number of erythroid cell of WT (n = 3) and thalassemic mice (n = 3). Erythroid differentiation of WT and thalassemic mice was determined by flow cytometry (B) and percentage of erythroid subpopulation (C). The morphology of erythroid differentiation of WT and thalassemic mice was examined by Wright-Giemsa staining (D). (E) Quantitative RT-PCR of miR-155 was determined in WT, thalassemic mice normalized by the expression of snoRNA202 (2−ΔCt). The experiments were carried out in triplicate. *p < 0.05, **p < 0.01 by Mann–Whitney U-Test.

The expression level of miR-155 during in vitro erythropoiesis was significantly increased in β-thalassemic mice in differentiation phase, day 1 and day 2, compared that of wild type mice (Fig. 2E). Our data suggest that increased proliferation and differentiation of thalassemic mouse erythroblasts, in both in vivo and in vitro erythropoiesis, may be associated with miR-155 upregulation.

The effects of miR-155 expression on erythroid proliferation and differentiation

To evaluate the functional effects of miR-155 expression on erythroid proliferation and differentiation, gain-and loss-of-function was investigated. Transfection with miRNA-mimic significantly elevated miR-155 expression in both β-thalassemic and WT mice, compared with untreated and negative control groups (p = 0.001) (Fig. 3A). The upregulation of miR-155 in β-thalassemic mice correlated with significantly decreased the percentage of erythroblast representing proerythroblast (region S1). A significant increase in the percentage of basophilic erythroblasts (region S2) was observed with a gain in miR-155 expression. However, increased miR-155 levels did not affect erythroid differentiation in the WT group (Figs. 3C and 3D).

Figure 3 Functional effects of miR-155 expression in presence of miR-155 -mimic and anti-miR-155 inhibitor.

Quantitative RT-PCR of miR-155 was determined in WT, thalassemic mice after transfection with miRNA-mimic (A) and anti-miR-155 inhibitor (B) in comparison between untreated, negative control and miRNA-mimic or anti-miR-155 inhibitor. Erythroid differentiation of WT and thalassemic mice was determined by flow cytometry after transfection with miRNA-mimic (C) and anti-miR-155 inhibitor (E) and percentage of erythroid subpopulation after transfection with miRNA-mimic (D) and anti-miR-155 inhibitor (F) in comparison between untreated, negative control and miRNA-mimic or anti-miR-155 inhibitor. The experiments were carried out in triplicate. *p < 0.05, **p < 0.01 by Mann–Whitney U-Test.

Loss-of-function during erythroid differentiation was assessed using an anti-miR-155 inhibitor. The results exhibited that transfection of cells with an anti-miR-155 inhibitor resulted in significant reduction in miR-155 expression compared to the untreated and negative control in both β-thalassemic (p = 0.05) and WT mice (p = 0.046) (Fig. 3B). In wild-type mice, the loss of miR-155 expression did not affect erythroid differentiation. However, a significant increase the percentage of early erythroblast (region S1) but decreased of the percentage of late erythroblast (region S2 and S3) was detected in β-thalassemic mice which were transfected with anti-miR-155 inhibitor, compared to that in untreated and negative inhibitor control-treated cells (Figs. 3E and 3F).

C-myc is a target of miR-155 in β-thalassemia erythropoiesis

To elucidate the probable functional association between miR-155 and cell differentiation, the predicted mRNA target genes of miR-155 were analyzed using TargetScan, MiRTarBase, and PicTar (Riolo et al., 2020; Undi, Kandi & Gutti, 2013). Three candidate mRNA targets, c-myc, bach-1, and pu-1, were selected from mRNA target prediction and previous publications showing their relationship with cell differentiation (Edalati Fathabad et al., 2017; Georgantas 3rd et al., 2007; Li et al., 2023; Palma et al., 2014; Penglong et al., 2023; Srinoun et al., 2017). Bach1 was selected from target of miR-155 in macrophage cells achieved from the bone marrow and spleen of β-thalassemic mice (Penglong et al., 2023). The level of c-myc, pu-1, and bach1 gene expression was determined in erythroblast cell derived from β-thalassemic and WT mice which were transfected with miR-155-mimic and -inhibitor. In the β-thalassemic and WT mice cells transfected with miR-155-mimic, the upregulation of miR-155 significantly decreased the expression of c-myc mRNA compared to that in untreated and negative miR-mimic-treated cells (p = 0.05) (Fig. 4A). However, there was no statistically significant difference in bach1 expression levels among the three groups in β-thalassemic and WT mice (p = 0.127 and p = 0.827, respectively) (Fig. 4B). The upregulation of miR-155 expression was significant increased pu-1 mRNA (p = 0.05) (Fig. 4C). Conversely, miR-155 downregulation correlated with the significant increase in c-myc mRNA expression in β-thalassemic and WT mice (p = 0.05) (Fig. 5A). Similar to the gain of miR-155 expression condition, there was no change in bach1 expression and also observed in pu-1 expression after anti-miR-155 inhibitor transfection in WT mice (p = 0.487 and p = 0.127, respectively) (Fig. 5B); however, the expression of pu-1 increased with miR-155 downregulation in β-thalassemic mice (p = 0.05) (Fig. 5C). Taken together, these results indicated that c-myc is a valid target gene of miR-155 that regulates erythroid differentiation.

Figure 4 Expression of c-myc in presence of miR-155-mimic.

Quantitative RT-PCR of target mRNA of miR-155, (A) c-myc, (B) bach1 and (C) pu-1 was determined in WT, thalassemic mice after transfection with miRNA-mimic in comparison between untreated, negative control and miRNA-mimic. This data was normalized by the expression of β-actin (Actb) (2−ΔΔCt). The experiments were carried out in triplicate. *p < 0.05, **p < 0.01 by Mann–Whitney U-Test.

Figure 5 Expression of c-myc in presence of anti-miR-155-inhibitor.

Quantitative RT-PCR of target mRNA of miR-155, (A) c-myc, (B) bach1 and (C) pu-1 was determined in WT, thalassemic mice after transfection with anti-miR-155 inhibitor in comparison between untreated, negative control and anti-miR-155 inhibitor. This data was normalized by the expression of β-actin (Actb) (2−ΔΔCt). The experiments were carried out in triplicate. *p < 0.05 by Mann–Whitney U-Test.

Discussion

IE in β-thalassemia is characterized by increased proliferation and expansion of immature erythroblasts, accelerated erythroid proliferation, maturation blockade at the polychromatophilic stage, and death of erythroid precursors (Pootrakul et al., 2000; Rund & Rachmilewitz, 2005). In this study, we focused on miR-155, which is involved in normal and IE in MDS (Bruchova et al., 2007; Das et al., 2021; Georgantas 3rd et al., 2007; Li et al., 2023; Masaki et al., 2007; Wan et al., 2020). Though miR-155 is associated with IE in MDS, the role of miR-155 in IE characterization of β-thalassemia erythroblast has not been studied. Therefore, to elucidate the molecular mechanism of IE in β-thalassemia, we examined both, in vivo and in vitro expression of miR-155 during erythropoiesis in this study.

We performed the investigation with cells obtained from bone marrow of β-thalassemic mice representing βIVSII-654 background. The mice typically exhibited anemia, ineffective erythropoiesis and splenomegaly, which is a hallmark of β-thalassemia. The our previous study displayed increased numbers of basophilic and polychromatic erythroblasts and decreased orthochromatic erythroblast (Srinoun et al., 2009). In the in vivo experiment, miR-155 expression was upregulated in both early and late stages of thalassemic erythroid cells compared to that in the WT. In in vitro experiment, the expression of miR-155 in untreated thalassemic erythroid cells was significantly higher than that in WT cells. Our data derived from in vitro bone marrow of β-thalassemic mice culture, demonstrated increased basophilic and polychromatic erythroblasts. This result was consistent with that of bone marrow culture obtained from β-thalassemia major patients, which also had increased erythroid expansion and accelerated erythroid proliferation especially polychromatophilic erythroblast state (Mathias et al., 2000). Additionally, the two-phase culture of erythroid progenitor cells is similar to the erythropoiesis in vivo process arising from the erythroblastic island niche, which consists of a central macrophage surrounded by developing erythroblasts. Macrophage in the erythroblastic island niche supports and promotes erythroblast proliferation and differentiation (May & Forrester, 2020; Rhodes et al., 2008). Thus, we have shown that both our in vivo and in vitro erythropoiesis in cells derived from β-thalassemic mice recapitulates the IE associated with the disease. These results suggested that the miR-155 may regulate IE in β-thalassemia characterized by increased of proliferation and differentiation of erythroblasts.

The upregulation of miR-155 in β-thalassemia was firstly reported by Das et al. (2021). Our findings in β-thalassemic mice are similar that study explored. Furthermore, miR-155 expression was significantly higher in MDS, an IE-related blood disease (Wan et al., 2020).

We evaluated the role of miR-155 in the regulation of erythropoiesis using miR-155 gain- and loss-of-function experiments. Erythroblasts were transfected with an miR-155 mimic or miR-155 inhibitor. Interestingly, the overexpression of miR-155 promoted the proliferation of thalassemia erythroid cells, that was not observed in the WT. In contrast, the population of basophilic and polychromatic erythroblasts decreased in erythroblast cells transfected with the miR-155 inhibitor. These findings suggest that miR-155 positively regulates erythropoiesis, particularly in thalassemia.

In the present study, we identified a candidate mRNA target of miR-155 in erythropoiesis. It has been reported that the targets of miR-155 include c-myc, bach-1, and pu-1 that which are related to cell differentiation (Edalati Fathabad et al., 2017; Georgantas 3rd et al., 2007; Li et al., 2023; Palma et al., 2014; Penglong et al., 2023; Srinoun et al., 2017). Bach1 was also selected from the targets of miR-155 in macrophage cells obtained from the bone marrow and spleen of β-thalassemic mice (Penglong et al., 2023). Our findings showed a downregulation of c-myc target gene expression in β-thalassemic erythroblast, and an association of erythroid differentiation with the downregulation of miR-155 expression. Overexpression of miR-155 contributes to inhibition of c-myc mRNA expression. In our study we found c-myc mRNA expression to be upregulated in cells transfected with miR-155 inhibitor. A previous study demonstrated that the upregulation of miR-155 regulates c-myc reduction in human FLT3-wildtype acute myeloid leukemia (AML) (Palma et al., 2014). Moreover, c-myc a target of miR-155, has been reported in human gastric carcinoma cells and confirmed that c-Myc is a target of miR-155 using a luciferase assay. They also observed post-translational suppression of c-Myc by miR-155, as shown by western blotting, with decreased protein levels of c-Myc upon treatment with miR-155 mimic. These results are consistent with our findings. However, the confirmation of protein levels for c-Myc, Bach-1, and PU.1 will be addressed in future studies (Sun et al., 2014).

The c-Myc proto-oncogene product is a transcription factor that plays an important role in regulating cell proliferation, differentiation, growth, and apoptosis (Dang et al., 2006; Hoffman & Liebermann, 2008; Iritani & Eisenman, 1999). In addition, c-Myc is critical for several aspects of hematopoietic development and function (Hoffman et al., 2002). In erythropoiesis, c-Myc protein is highly expressed in early erythroid progenitors, but rapidly decreases in the late stage of erythroid cells, suggesting that loss of c-Myc might be required for normal maturation of erythroid progenitors (Goupille et al., 2012). Jayapal et al. demonstrated that the protein level of c-Myc was significantly reduced concurring with cell cycle arrest in G1 phase during late-stage erythroid maturation. When c-Myc overexpressed, the gene expression analysis revealed the up-regulation of several positive regulators of G1-S cell cycle checkpoint. They also showed that c-Myc up-regulated the histone acetyltransferase Gcn5, and ectopic Gcn5 expression partially blocked enucleation and inhibited the late stage erythroid nuclear condensation and histone deacetylation (Jayapal et al., 2010). This result was consistent with the upregulation of c-myc mRNA expression in thalassemic erythroblasts transfected with an miR-155 inhibitor. These results reveal the importance of miR-155 in IE which may regulate c-myc that control proliferation and terminal maturation of erythroblasts in β-thalassemia. The limitations of the present study include the limited number of mice (n = 3) in each group, and the lack of outcomes on examine whether miR-155 directly inhibits c-myc mRNA by luciferase. In addition, the expression of c-myc at the protein level should be performed in future studies.

Conclusions

In this study, we propose a role of miR-155 in IE in β-thalassemia. Our study showed that miR-155 expression was upregulated in thalassemic mice; in both in vivo and in vitro experiments; corresponding to increase in proliferation and differentiation of erythroblasts in β-thalassemic mice. Additionally, we clarified the possible functional connection between miR-155 and erythroid cell differentiation via c-myc expression. Although the role of miR-155 in regulating c-myc in IE was recapitulated in β-thalassemia especially the increased proliferation and accelerated erythroid differentiation, further complementary studies are required to address some more aspects, including in terminal maturation in β-thalassemia.

Supplemental Information

Supplemental Information 1 Author checklist

Supplemental Information 2 MIQE

Data S1 Raw data

We thank the Thalassemia Research Center, Institute of Molecular Biosciences, Mahidol.

Additional Information and Declarations

Competing Interests

Author Contributions

Animal Ethics

Ethics

Data Availability

The authors declare there are no competing interests.

Tipparat Penglong conceived and designed the experiments, analyzed the data, prepared figures and/or tables, authored or reviewed drafts of the article, and approved the final draft.

Nuttanan Pholngam performed the experiments, prepared figures and/or tables, authored or reviewed drafts of the article, and approved the final draft.

Nasra Tehyoh performed the experiments, prepared figures and/or tables, authored or reviewed drafts of the article, and approved the final draft.

Natta Tansila analyzed the data, authored or reviewed drafts of the article, and approved the final draft.

Hansuk Buncherd analyzed the data, authored or reviewed drafts of the article, and approved the final draft.

Supinya Thanapongpichat analyzed the data, authored or reviewed drafts of the article, and approved the final draft.

Kanitta Srinoun conceived and designed the experiments, performed the experiments, analyzed the data, prepared figures and/or tables, authored or reviewed drafts of the article, and approved the final draft.

The following information was supplied relating to ethical approvals (i.e., approving body and any reference numbers):

The Institute of Science and Technology for Research and Development and the Institute of Molecular Biosciences, Mahidol University, Institute Animal Care and Use Committee provided full approval for this research (approval number MUSTA 2008-004 and COA.NO.IMB-ACUC 2021/011).

The following information was supplied relating to ethical approvals (i.e., approving body and any reference numbers):

Institute of Molecular Biosciences, Mahidol University granted ethical approval to carry out the study within its facilities (Ethical Application Ref: MUSTA 2008-004 and COA.NO.IMB-ACUC 2021/011)

The following information was supplied regarding data availability:

The data is available at Flow Repository: FR-FCM-Z7DR.

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
