# Peer review of "Expression of microRNA-155 in thalassemic erythropoiesis"

_PeerJ, doi:10.7717/peerj.18054_

## Round 0.1 · original submission · Major Revisions

Dear Dr. Srinoun,

If you feel you can revise your manuscript according to the reviewers' comments, please revise your manuscript and submit it. Please also send us your written responses to each of the reviewers' comments.

Yours,

Yoshi

Prof. Yoshinori Marunaka, M.D., Ph.D.

Reviewer 1 ·

Basic reporting

No comments

Experimental design

1. Were there any specific reasons for choosing male and female (8-12) week-old mice?

2. Please provide the exact p-value for comparison between different treatment groups such as miR155 mimic, anti-miR-155 inhibitor, untreated and negative miR-mimic treated cells for each gene expression level (c-myc, bach1 and pu1).

3. Has the power analysis conducted for this study to ensure the statistical power to detect differences in gene expression?

Validity of the findings

No comments

Additional comments

-

·

Basic reporting

Giemsa staining images should have legible scale bars to clarify the magnification.

Experimental design

The article investigates the role of miR-155 in the ineffective erythropoiesis seen in β-thalassemia, a condition characterized by an imbalance in erythroid cell proliferation and differentiation, resulting in anemia. It is found that miR-155, a microRNA involved in hematopoiesis, is upregulated in β-thalassemia. The study shows that miR-155 influences erythroblast proliferation and differentiation, with increased levels leading to an accumulation of immature erythroblasts. It identifies c-myc as a significant target gene of miR-155, crucial for regulating erythroid differentiation. Overall, the article is compelling. However, addressing the following minor comment will strengthen the publication:

The author suggests a possible functional connection between miR-155 and erythroid cell differentiation via c-myc expression. Although further studies are recommended, it is essential to at least discuss the genes identified in Jayapal et al., 2010.

Validity of the findings

The relative expression of c-myc, bach-1, and pu-1 should also be tested using western blotting to validate the findings.

Reviewer 3 ·

Basic reporting

The study aimed to investigate miR-155 in B-thal mice. The strengths of the study is the use of an appropriate mouse model and the conceptualization. However, there were many weaknesses in the results and approach sections. Effect of miR-155 appeared modest and functional assays, such as luciferase assay confirming the effect of miR-155 binding to c-myc is needed. There were also grammatical errors throughout the manuscript and several figures were very poor quality. Major revisions are needed

Line 77 “diseases severity” …. Remove “s” from diseases. It should be disease severity
Please clarify what is meant at lines 161 and 162 where it states (n=3/genotype)
Figure 1A and B should also show what occurs in WT mice for comparison
Incomplete sentence at lines 215 – 216 … miRNA expression determined using qRT-PCR revealed that in WT mice and b-thalassemic mice.
Results in figure 1 to show miR-155 expression level in early erythroblast was significantly lower than that of late erythroblast in b-thalassemic mice are modest at ranges from 0.02 – 0.08
The methods state that snoRNA202 was used as a reference control yet Figure 1C shows normalization to let7. Please clarify
An explanation as to why differentiation was done for only Days 0 – 2 is needed.
Please revise Line 246 to state …Expression of miR-155 is involved in erythroid proliferation and differentiation.
Proofread manuscript for grammatical errors and additional words or lack thereof
Figures 4 and 5 are poor quality and not clear enough to interpret.
The authors need to conduct more functional analysis, besides rt-pcr, to confirm that c-myc is a target of miR-155.

Experimental design

See above

Validity of the findings

See above

---

## Round 0.2 · accepted · Accept

Congratulations again, and thank you for your submission.
Yours,
Yoshi
Prof. Yoshinori Marunaka, M.D., Ph.D
Academic Editor
PeerJ Life & Environment

Reviewer 1 ·

Basic reporting

No comment

Experimental design

No comment

Validity of the findings

No comment

Additional comments

The comments have been addressed satisfactorily